# Extreme Classification via Adversarial Softmax Approximation

**Robert Bamler & Stephan Mandt**
Department of Computer Science
University of California, Irvine
{rbamler,mandt}@uci.edu

## Abstract

Training a classifier over a large number of classes, known as 'extreme classification', has become a topic of major interest with applications in technology, science, and e-commerce. Traditional softmax regression induces a gradient cost proportional to the number of classes $C$, which often is prohibitively expensive. A popular scalable softmax approximation relies on uniform negative sampling, which suffers from slow convergence due a poor signal-to-noise ratio. In this paper, we propose a simple training method for drastically enhancing the gradient signal by drawing negative samples from an adversarial model that mimics the data distribution. Our contributions are three-fold: (i) an adversarial sampling mechanism that produces negative samples at a cost only logarithmic in $C$, thus still resulting in cheap gradient updates; (ii) a mathematical proof that this adversarial sampling minimizes the gradient variance while any bias due to non-uniform sampling can be removed; (iii) experimental results on large scale data sets that show a reduction of the training time by an order of magnitude relative to several competitive baselines.

## 1 Introduction

In many problems in science, healthcare, or e-commerce, one is interested in training classifiers over an enormous number of classes: a problem known as 'extreme classification' (Agrawal et al., 2013; Jain et al., 2016; Prabhu & Varma, 2014; Siblini et al., 2018). For softmax (aka multinomial) regression, each gradient step incurs a cost proportional to the number of classes $C$. As this may be prohibitively expensive for large $C$, recent research has explored more scalable softmax approximations which circumvent the linear scaling in $C$. Progress in accelerating the training procedure and thereby scaling up extreme classification promises to dramatically improve, e.g., advertising (Prabhu et al., 2018), recommender systems, ranking algorithms (Bhatia et al., 2015; Jain et al., 2016), and medical diagnostics (Bengio et al., 2019; Lippert et al., 2017; Baumel et al., 2018)

While scalable softmax approximations have been proposed, each one has its drawbacks. The most popular approach due to its simplicity is 'negative sampling' (Mnih & Hinton, 2009; Mikolov et al., 2013), which turns the problem into a binary classification between so-called 'positive samples' from the data set and 'negative samples' that are drawn at random from some (usually uniform) distribution over the class labels. While negative sampling makes the updates cheaper since computing the gradient no longer scales with $C$, it induces additional gradient noise that leads to a poor signal-to-noise ratio of the stochastic gradient estimate. Improving the signal-to-noise ratio in negative sampling while still enabling cheap gradients would dramatically enhance the speed of convergence.

In this paper, we present an algorithm that inherits the cheap gradient updates from negative sampling while still preserving much of the gradient signal of the original softmax regression problem. Our approach rests on the insight that the signal-to-noise ratio in negative sampling is poor since there is no association between input features and their artificial labels. If negative samples were harder to discriminate from positive ones, a learning algorithm would obtain a better gradient signal close to the optimum. Here, we make these arguments mathematically rigorous and propose a non-uniform sampling scheme for scalably approximating a softmax classification scheme. Instead of sampling labels uniformly, our algorithm uses an *adversarial auxiliary model* to draw 'fake' labels that are more realistic by taking the input features of the data into account. We prove that such procedure

reduces the gradient noise of the algorithm, and in fact minimizes the gradient variance in the limit where the auxiliary model optimally mimics the data distribution.

A useful adversarial model should require only little overhead to be fitted to the data, and it needs to be able to *generate* negative samples quickly in order to enable inexpensive gradient updates. We propose a probabilistic version of a decision tree that has these properties. As a side result of our approach, we show how such an auxiliary model can be constructed and efficiently trained. Since it is almost hyperparameter-free, it does not cause extra complications when tuning models.

The final problem that we tackle is to remove the bias that the auxiliary model causes relative to our original softmax classification. Negative sampling is typically described as a softmax approximation; however, only *uniform* negative sampling correctly approximates the softmax. In this paper, we show that the bias due to non-uniform negative sampling can be easily removed at test time.

The stucture of our paper reflects our main contributions as follows:

1.  We present a new scalable softmax approximation (Section 2). We show that non-uniform sampling from an auxiliary model can improve the signal-to-noise ratio. The best performance is achieved when this sampling mechanism is *adversarial*, i.e., when it generates fake labels that are hard to discriminate from the true ones. To allow for efficient training, such adversarial samples need to be generated at a rate sublinear (e.g., logarithmic) in $C$.

2.  We design a new, simple adversarial auxiliary model that satisfies the above requirements (Section 3). The model is based on a probabilistic version of a decision tree. It can be efficiently pre-trained and included into our approach, and requires only minimal tuning.

3.  We present mathematical proofs that (i) the best signal-to-noise ratio in the gradient is obtained if the auxiliary model best reflects the true dependencies between input features and labels, and that (ii) the involved bias to the softmax approximation can be exactly quantified and cheaply removed at test time (Section 4).

4.  We present experiments on two classification data sets that show that our method outperforms all baselines by at least one order of magnitude in training speed (Section 5).

We discuss related work in Section 6 and summarize our approach in Section 7.

## 2 AN ADVERSARIAL SOFTMAX APPROXIMATION

We propose an efficient algorithm to train a classifier over a large set of classes, using an asymptotic equivalence between softmax classification and negative sampling (Subsection 2.1). To speed up convergence, we generalize this equivalence to model-based negative sampling in Subsection 2.2.

### 2.1 ASYMPTOTIC EQUIVALENCE OF SOFTMAX CLASSIFICATION AND NEGATIVE SAMPLING

**Softmax Classification (Notation).** We consider a training data set $\mathcal{D} = \{(x_i, y_i)\}_{i=1:N}$ of $N$ data points with $K$-dimensional feature vectors $x_i \in \mathbb{R}^K$. Each data point has a single label $y_i \in \mathcal{Y}$ from a discrete label set $\mathcal{Y}$. A softmax classifier is defined by a set of functions $\{\xi_y\}_{y \in \mathcal{Y}}$ that map a feature vector $x$ and model parameters $\theta$ to a score $\xi_y(x, \theta) \in \mathbb{R}$ for each label $y$. Its loss function is

$$\ell_{\text{softmax}}(\theta) = \sum_{(x,y) \in \mathcal{D}} \left[ -\xi_y(x, \theta) + \log \left( \sum_{y' \in \mathcal{Y}} e^{\xi_{y'}(x,\theta)} \right) \right]. \tag{1}$$

While the first term encourages high scores $\xi_y(x, \theta)$ for the correct labels $y$, the second term encourages low scores for all labels $y' \in \mathcal{Y}$, thus preventing degenerate solutions that set all scores to infinity. Unfortunately, the sum over $y' \in \mathcal{Y}$ makes gradient based minimization of $\ell_{\text{softmax}}(\theta)$ expensive if the label set $\mathcal{Y}$ is large. Assuming that evaluating a single score $\xi_{y'}(x, \theta)$ takes $O(K)$ time, each gradient step costs $O(KC)$, where $C = |\mathcal{Y}|$ is the size of the label set.

**Negative Sampling.** Negative sampling turns classification over a large label set $\mathcal{Y}$ into binary classification between so-called positive and negative samples. One draws positive samples $(x, y)$

from the training set and constructs negative samples $(x, y')$ by drawing random labels $y'$ from some noise distribution $p_n$. One then trains a logistic regression by minimizing the stochastic loss function

$$\ell_{\text{neg.sampl.}}(\phi) = \sum_{(x,y) \in \mathcal{D}} \left[ -\log \sigma(\xi_y(x, \phi)) - \log \sigma(-\xi_{y'}(x, \phi)) \right] \qquad \text{where} \qquad y' \sim p_n \quad (2)$$

with the sigmoid function $\sigma(z) = 1/(1 + e^{-z})$. Here, we used the same score functions $\xi_y$ as in Eq. 1 but introduced different model parameters $\phi$ so that we can distinguish the two models. Gradient steps for $\ell_{\text{neg.sampl}}(\phi)$ cost only $O(K)$ time as there is no sum over all labels $y' \in \mathcal{Y}$.

**Asymptotic Equivalence.** The models in Eqs. 1 and Eq. 2 are exactly equivalent in the *non-parametric limit*, i.e., if the function class $x \mapsto \xi_y(x, \theta)$ is flexible enough to map $x$ to any possible score. A further requirement is that $p_n$ in Eq. 2 is the uniform distribution over $\mathcal{Y}$. If both conditions hold, it follows that if $\theta^*$ and $\phi^*$ minimize Eq. 1 and Eq. 2, respectively, they learn identical scores,

$$\xi_y(x, \theta^*) = \xi_y(x, \phi^*) + \text{const.} \qquad \text{(for uniform } p_n). \tag{3}$$

As a consequence, one is free to choose the loss function that is easier to minimize. While gradient steps are cheaper by a factor of $O(C)$ for negative sampling, the randomly drawn negative samples increase the variance of the stochastic gradient estimator and worsen the signal-to-noise ratio of the gradient, slowing-down convergence. The next section combines the strengths of both approaches.

## 2.2 ADVERSARIAL NEGATIVE SAMPLING

**Overview.** We propose a generalized variant of negative sampling that reduces the gradient noise. The main idea is to train with negative samples $y'$ that are hard to distinguish from positive samples. We draw $y'$ from a conditional noise distribution $p_n(y'|x)$ using an auxiliary model. This introduces a bias, which we remove at prediction time. In summary our proposed approach consists of three steps:

1. Parameterize the noise distribution $p_n(y'|x)$ by an auxiliary model and fit it to the data set.
2. Train a classifier via negative sampling (Eq. 2) using adversarial negative samples from the auxiliary model fitted in Step 1 above. For our proposed auxiliary model, drawing a negative sample costs only $O(k \log C)$ time with some $k < K$, i.e., it is sublinear in $C$.
3. The resulting model has a bias. When making predictions, remove the bias by mapping it to an unbiased softmax classifier using the generalized asymptotic equivalence in Eq. 5 below.

We elaborate on the above Step 1 in Section 3. In the present section, we focus instead on Step 2 and its dependency on the choice of noise distribution $p_n$, and on the bias removal (Step 3).

**Why Adversarial Noise Improves Learning.** We first provide some intuition why uniform negative sampling is not optimal, and how sampling from a non-uniform noise distribution may improve the gradient signal. We argue that the poor gradient signal is caused by the fact that negative samples are too easy to distinguish from positive samples. A data set with many categories is typically comprised of several hierarchical clusters, with large clusters of generic concepts and small sub-clusters of specialized concepts. When drawing negative samples uniformly across the data set, the correct label will likely belong to a different generic concept than the negative sample. For example, an image classifier will therefore quickly learn to distinguish, e.g., dogs from bicycles, but since negative samples from the same cluster are rare, it takes much longer to learn the differences between a Boston Terrier and a French Bulldog. The model quickly learns to assign very low scores $\xi_{y'}(x, \phi) \ll 0$ to such 'obviously wrong' labels, making their contribution to the gradient exponentially small,

$$||\nabla_\phi \log \sigma(-\xi_{y'}(x, \phi))||_2 = \sigma(\xi_{y'}(x, \phi)) \, ||\nabla_\phi \xi_{y'}(x, \phi)||_2$$
$$\approx e^{\xi_{y'}(x, \phi)} \, ||\nabla_\phi \xi_{y'}(x, \phi)||_2 \qquad \text{for } \xi_{y'}(x, \phi) \ll 0. \tag{4}$$

A similar vanishing gradient problem was pointed out for word embeddings by Chen et al. (2018). Here, the vanishing gradient is due to different word frequencies, and a popular solution is therefore to draw negative samples from a nonuniform but unconditional noise distribution $p_n(y')$ based on the empirical word frequencies (Mikolov et al., 2013). This introduces a bias which does not matter for word embeddings since the focus is not on classification but rather on learning useful representations.

Going beyond frequency-adjusted negative sampling, we show that one can drastically improve the procedure by generating negative samples from an auxiliary model. We therefore propose to generate negative samples $y' \sim p_{\mathrm{n}}(y'|x)$ conditioned on the input feature $x$. This has the advantage that the distribution of negative samples can be made much more similar to the distribution of positive samples, leading to a better signal-to-noise ratio. One consequence is that the introduced bias can no longer be ignored, which is what we address next.

**Bias Removal.** Negative sampling with a nonuniform noise distribution introduces a bias. For a given input feature vector $x$, labels $y'$ with a high noise probability $p_{\mathrm{n}}(y'|x)$ are frequently drawn as negative samples, causing the model to learn a low score $\xi_{y'}(x, \phi^*)$. Conversely, a low $p_{\mathrm{n}}(y'|x)$ leads to an inflated score $\xi_{y'}(x, \phi^*)$. It turns out that this bias can be easily quantified via a generalization of Eq. 3. We prove in Theorem 1 (Section 4) that in the nonparametric limit for arbitrary $p_{\mathrm{n}}(y'|x)$,

$$\xi_y(x, \theta^*) = \xi_y(x, \phi^*) + \log p_{\mathrm{n}}(y|x) + \mathrm{const}. \qquad \text{(nonparametric limit and arbitrary } p_{\mathrm{n}}). \qquad (5)$$

Eq. 5 is an asymptotic equivalence between softmax classification (Eq. 1) and generalized negative sampling (Eq. 2). While strict equality holds only in the nonparametric limit, many models are flexible enough that Eq. 5 holds approximately in practice. Eq. 5 allows us to make unbiased predictions by mapping biased negative sampling scores $\xi_y(x, \phi^*)$ to unbiased softmax scores $\xi_y(x, \theta^*)$. There is no need to solve for the corresponding model parameters $\theta^*$, the scores $\xi_y(x, \theta^*)$ suffice for predictions.

**Regularization.** In practice, softmax classification typically requires a regularizer with some strength $\lambda > 0$ to prevent overfitting. With the asymptotic equivalence in Eq. 5, regularizing the softmax scores $\xi_y(x, \theta)$ is similar to regularizing $\xi_y(x, \phi) + \log p_{\mathrm{n}}(y|x)$ in the proposed generalized negative sampling method. We thus propose to use the following regularized variant of Eq. 2,

$$\ell_{\mathrm{neg.sampl.}}^{(\mathrm{reg.})}(\phi) = \frac{1}{N} \sum_{(x,y)\in\mathcal{D}} \Big[ -\log \sigma(\xi_y(x, \phi)) + \lambda\big(\xi_y(x, \phi) + \log p_{\mathrm{n}}(y|x)\big)^2 \qquad (6)$$
$$ -\log \sigma(-\xi_{y'}(x, \phi)) + \lambda\big(\xi_{y'}(x, \phi) + \log p_{\mathrm{n}}(y'|x)\big)^2 \Big]; \quad y' \sim p_{\mathrm{n}}(y'|x).$$

**Comparison to GANs.** The use of adversarial negative samples, i.e., negative samples that are designed to 'confuse' the logistic regression in Eq. 2, bears some resemblance to generative adversarial networks (GANs) (Goodfellow et al., 2014). The crucial difference is that GANs are *generative* models, whereas we train a *discriminative* model over a discrete label space $\mathcal{Y}$. The 'generator' $p_{\mathrm{n}}$ in our setup only needs to find a rough approximation of the (conditional) label distribution because the final predictive scores in Eq. 5 combine the 'generator scores' $\log p_{\mathrm{n}}(y|x)$ with the more expressive 'discriminator scores' $\xi_y(x, \phi^*)$. This allows us to use a very restrictive but efficient generator model (see Section 3 below) that we can keep constant while training the discriminator. By contrast, the focus in GANs is on finding the best possible generator, which requires concurrent training of a generator and a discriminator via a potentially unstable nested min-max optimization.

## 3 CONDITIONAL GENERATION OF ADVERSARIAL SAMPLES

Having proposed a general approach for improved negative sampling with an adversarial auxiliary model $p_n$ (Section 2), we now describe a simple construction for such a model that satisfies all requirements. The model is essentially a probabilistic version of a decision tree which is able to conditionally generate negative samples by ancestral sampling. Readers who prefer to proceed can skip this section without loosing the main thread of the paper.

Our auxiliary model has the following properties: (i) it can be efficiently fitted to the training data $\mathcal{D}$ requiring minimal hyperparameter tuning and subleading computational overhead over the training of the main model; (ii) drawing negative samples $y' \sim p_{\mathrm{n}}(y'|x)$ scales only as $O(\log |\mathcal{Y}|)$, thus improving over the linear scaling of the softmax loss function (Eq. 1); and (iii) the log likelihood $\log p_{\mathrm{n}}(y|x)$ can be evaluated explicitly so that we can apply the bias removal in Eq. 5. Satisfying requirements (i) and (ii) on model efficiency comes at the cost of some model performance. This is an acceptable trade-off since the performance of $p_{\mathrm{n}}$ affects only the quality of negative samples.

**Model.** Our auxiliary model for $p_{\mathrm{n}}$ is inspired by the Hierarchical Softmax model due to Morin & Bengio (2005). It is a balanced probabilistic binary decision tree, where each leaf node is mapped

uniquely to a label $y \in \mathcal{Y}$. A decision tree imposes a hierarchical structure on $\mathcal{Y}$, which can impede performance if it does not reflect any semantic structure in $\mathcal{Y}$. Morin & Bengio (2005) rely on an explicitly provided semantic hierarchical structure, or 'ontology'. Since an ontology is often not available, we instead construct a hierarchical structure in a data driven way. Our method has some similarity to the approach by Mnih & Hinton (2009), but it is more principled in that we fit both the model parameters and the hierarchical structure by maximizing a single log likelihood function.

To sample from the model, one walks from the tree's root to some leaf. At each node $\nu$, one makes a binary decision $\zeta \in \{\pm 1\}$ whether to continue to the right child ($\zeta = 1$) or to the left child ($\zeta = -1$). Given a feature vector $x$, we model the likelihood of these decisions as $\sigma\big(\zeta(w_\nu^\top x + b_\nu)\big)$, where the weight vector $w_\nu$ and the scalar bias $b_\nu$ are model parameters associated with node $\nu$. Denoting the unique path $\pi_y$ from the root node $\nu_0$ to the leaf node associated with label $y$ as a sequence of nodes and binary decisions, $\pi_y \equiv \big((\nu_0, \zeta_0), (\nu_1, \zeta_1), \dots\big)$, the log likelihood of the training set $\mathcal{D}$ is thus

$$\mathcal{L} := \sum_{(x,y) \in \mathcal{D}} \log p_{\mathrm{n}}(y|x) = \sum_{(x,y) \in \mathcal{D}} \left[ \sum_{(\nu,\zeta) \in \pi_y} \log \sigma\big(\zeta(w_\nu^\top x + b_\nu)\big) \right]. \tag{7}$$

**Greedy Model Fitting.** We maximize the likelihood $\mathcal{L}$ in Eq. 7 over (i) the model parameters $w_\nu$ and $b_\nu$ of all nodes $\nu$, and (ii) the hierarchical structure, i.e., the mapping between labels $y$ and leaf nodes. The latter involves an exponentially large search space, making exact maximization intractable. We use a greedy approximation where we recursively split the label set $\mathcal{Y}$ into halves and associate each node $\nu$ with a subset $\mathcal{Y}_\nu \subseteq \mathcal{Y}$. We start at the root node $\nu_0$ with $\mathcal{Y}_{\nu_0} = \mathcal{Y}$ and finishing at the leaves with a single label per leaf. For each node $\nu$, we maximize the terms in $\mathcal{L}$ that depend on $w_\nu$ and $b_\nu$. These terms correspond to data points with a label $y \in \mathcal{Y}_\nu$, leading to the objective

$$\mathcal{L}_\nu := \sum_{(x,y) \in \mathcal{D} \wedge y \in \mathcal{Y}_\nu} \log \sigma\big(\zeta_y(w_\nu^\top x + b_\nu)\big). \tag{8}$$

We alternate between a continuous maximization of $\mathcal{L}_\nu$ over $w_\nu$ and $b_\nu$, and a discrete maximization over the binary indicators $\zeta_y \in \{\pm 1\}$ that define how we split $\mathcal{Y}_\nu$ into two equally sized halves. The continuous optimization is over a convex function and it converges quickly to machine precision with Newton ascent, which is free of hyperparameters like learning rates. For the discrete optimization, we note that changing $\xi_y$ for any $y \in \mathcal{Y}_\nu$ from $-1$ to $1$ (or from $1$ to $-1$) increases (or decreases) $\mathcal{L}_\nu$ by

$$\Delta_y := \sum_{x \in \mathcal{D}_y} \big[\log \sigma(w_\nu^\top x + b_\nu) - \log \sigma(-w_\nu^\top x - b_\nu)\big] = \sum_{x \in \mathcal{D}_y} \big(w_\nu^\top x + b_\nu\big). \tag{9}$$

Here, the sums over $\mathcal{D}_y$ run over all data points in $\mathcal{D}$ with label $y$, and the second equality is an algebraic identity of the sigmoid function. We maximize $\mathcal{L}_\nu$ over all $\zeta_y$ under the boundary condition that the split be into equally sized halves by setting $\zeta_y \leftarrow 1$ for the half of $y \in \mathcal{Y}_\nu$ with largest $\Delta_y$ and $\zeta_y \leftarrow -1$ for the other half. If this changes any $\zeta_y$ then we switch back to the continuous optimization. Otherwise, we have reached a local optimum for node $\nu$, and we proceed to the next node.

**Technical Details.** In the interest of clarity, the above description left out the following details. Most importantly, to prioritize efficiency over accuracy, we preprocess the feature vectors $x$ and project them to a smaller space $\mathbb{R}^k$ with $k < K$ using principal component analysis (PCA). Sampling from $p_{\mathrm{n}}$ thus costs only $O(k \log |\mathcal{Y}|)$ time. This dimensionality reduction only affects the quality of negative samples. The main model (Eq. 2) still operates on the full feature space $\mathbb{R}^K$. Second, we add a quadratic regularizer $-\lambda_{\mathrm{n}}(||w_\nu||^2 + b_\nu^2)$ to $\mathcal{L}_\nu$, with strength $\lambda_{\mathrm{n}}$ set by cross validation. Third, we introduce uninhabited padding labels if $|\mathcal{Y}|$ is not a power of two. We ensure that $p_{\mathrm{n}}(\tilde{y}|x) = 0$ for all padding labels $\tilde{y}$ by setting $b_\nu$ to a very large positive or negative value if either of $\nu$'s children contains only padding labels. Finally, we initialize the optimization with $b_\nu \leftarrow 0$ and by setting $w_\nu \in \mathbb{R}^k$ to the dominant eigenvector of the covariance matrix of the set of vectors $\{\sum_{x \in \mathcal{D}_y} x\}_{y \in \mathcal{Y}_\nu}$.

## 4 THEORETICAL ASPECTS

We formalize and prove the two main premises of the algorithm proposed in Section 2.2. Theorem 1 below states the equivalence between softmax classification and negative sampling (Eq. 5), and Theorem 2 formalizes the claim that adversarial negative samples maximize the signal-to-noise ratio.

**Theorem 1.** *In the nonparametric limit, the optimal model parameters $\theta^*$ and $\phi^*$ that minimize $\ell_{softmax}(\theta)$ in Eq. 1 and $\ell_{neg.sampl.}(\phi)$ in Eq. 2, respectively, satisfy Eq. 5 for all $x$ in the data set and all $y \in \mathcal{Y}$. Here, the "const." term on the right-hand side of Eq. 5 is independent of $y$.*

*Proof.* Minimizing $\ell_{\text{softmax}}(\theta)$ fits the maximum likelihood estimate of a model with likelihood $p_\theta(y|x) = e^{\xi_y(x,\theta)}/Z_\theta(x)$ with normalization $Z_\theta(x) = \sum_{y' \in \mathcal{Y}} e^{\xi_{y'}(x,\theta)}$. In the nonparametric limit, the score functions $\xi_y(x,\theta)$ are arbitrarily flexible, allowing for a perfect fit, thus

$$p_\mathcal{D}(y|x) = p_{\theta^*}(y|x) = e^{\xi_y(x,\theta^*)}/Z_{\theta^*}(x) \qquad \text{(nonparametric limit).} \qquad (10)$$

Similarly, $\ell_{\text{neg.sampl.}}(\phi)$ is the maximum likelihood objective of a binomial model that discriminates positive from negative samples. The nonparametric limit admits again a perfect fit so that the learned ratio of positive rate $\sigma(\xi_y(x,\phi))$ to negative rate $\sigma(-\xi_y(x,\phi))$ equals the empirical ratio,

$$\frac{p_\mathcal{D}(y|x)}{p_\text{n}(y|x)} = \frac{\sigma(\xi_y(x,\phi^*))}{\sigma(-\xi_y(x,\phi^*))} = e^{\xi_y(x,\phi^*)} \qquad \text{(nonparametric limit)} \qquad (11)$$

where we used the identity $\sigma(z)/\sigma(-z) = e^z$. Inserting Eq. 10 for $p_\mathcal{D}(y|x)$ and taking the logarithm leads to Eq. 5. Here, the "const." term works out to $\log Z_{\theta^*}(x)$, which is indeed independent of $y$. $\quad\square$

**Signal-to-Noise Ratio.** In preparation for Theorem 2 below, we define a quantitative measure for the signal-to-noise ratio (SNR) in stochastic gradient descent (SGD). In the vicinity of the minimum $\phi^*$ of a loss function $\ell(\phi)$, the gradient $g \approx H_\ell(\phi - \phi^*)$ is approximately proportional to the Hessian $H_\ell$ of $\ell$ at $\phi^*$. SGD estimates $g$ via stochastic gradient estimates $\hat{g}$, whose noise is measured by the covariance matrix $\text{Cov}[\hat{g}, \hat{g}]$. Thus, the eigenvalues $\{\eta_i\}$ of the matrix $A := H_\ell \text{Cov}[\hat{g}, \hat{g}]^{-1}$ measure the SNR along different directions in parameter space. We define an overall scalar SNR $\bar{\eta}$ as

$$\bar{\eta} := \frac{1}{\sum_i 1/\eta_i} = \frac{1}{\text{Tr}[A^{-1}]} = \frac{1}{\text{Tr}[\text{Cov}[\hat{g},\hat{g}]\,H_\ell^{-1}]}. \qquad (12)$$

Here, we sum over the inverses $1/\eta_i$ rather than $\eta_i$ so that $\bar{\eta} \leq \min_i \eta_i$ and thus maximizing $\bar{\eta}$ encourages large values for all $\eta_i$. The definition in Eq. 12 has the useful property that $\bar{\eta}$ is invariant under arbitrary invertible reparameterization of $\phi$. Expressing $\phi$ in terms of new model parameters $\phi'$ maps $H_\ell$ to $J^\top H_\ell J$ and $\text{Cov}[\hat{g}, \hat{g}]$ to $J^\top \text{Cov}[\hat{g}, \hat{g}]J$, where $J := \partial\phi/\partial\phi'$ is the Jacobian. Inserting into Eq. 12 and using the cyclic property of the trace, $\text{Tr}[PQ] = \text{Tr}[QP]$, all Jacobians cancel.

**Theorem 2.** *For negative sampling (Eq. 2) in the nonparametric limit, the signal-to-noise ratio $\bar{\eta}$ defined in Eq. 12 is maximal if $p_n = p_\mathcal{D}$, i.e., for adversarial negative samples.*

*Proof.* In the nonparametric limit, the scores $\xi_y(x,\phi)$ can be regarded as independent variables for all $x$ and $y$. We therefore treat the scores directly as model parameters, using the invariance of $\bar{\eta}$ under reparameterization. Using only Eq. 2, Eq. 11, and properties of the $\sigma$-function, we show in Appendix A.1 that the Hessian of the loss function is diagonal in this coordinate system, and given by

$$H_\ell = \text{diag}(\alpha_{x,y}) \qquad \text{with} \qquad \alpha_{x,y} = p_\text{n}(y|x)\,\sigma(\xi_y(x,\phi^*)) \qquad (13)$$

and that the noise covariance matrix is block diagonal,

$$\text{Cov}[\hat{g},\hat{g}] = \text{diag}(C_x) \qquad \text{with blocks} \qquad C_x = N\,\text{diag}(\alpha_{x,:}) - 2N\alpha_{x,:}\alpha_{x,:}^\top \qquad (14)$$

where $\alpha_{x,:} \equiv (\alpha_{x,y})_{y \in \mathcal{Y}}$ denotes a $|\mathcal{Y}|$-dimensional column vector. Thus, the trace in Eq. 12 is

$$\frac{1}{\bar{\eta}} = \sum_x \text{Tr}\left[C_x \,\text{diag}\left(\frac{1}{\alpha_{x,:}}\right)\right] = N\sum_x \text{Tr}\left[I - 2\alpha_{x,:}\alpha_{x,:}^\top \,\text{diag}\left(\frac{1}{\alpha_{x,:}}\right)\right] = N\sum_x \left[|\mathcal{Y}| - 2\sum_{y \in \mathcal{Y}} \alpha_{x,y}\right]. \qquad (15)$$

We thus have to maximize $\sum_{y \in \mathcal{Y}} \alpha_{x,y}$ for each $x$ in the training set. We find from Eq. 13 and Eq. 11,

$$\sum_{y \in \mathcal{Y}} \alpha_{x,y} \overset{(13)}{=} \mathbb{E}_{p_\text{n}(y|x)}\left[\sigma(\xi_y(x,\phi^*))\right] = \mathbb{E}_{p_\text{n}(y|x)}\left[\frac{1}{1+e^{-\xi_y(x,\phi^*)}}\right] \overset{(11)}{=} \mathbb{E}_{p_\text{n}(y|x)}\left[f\left(\frac{p_\mathcal{D}(y|x)}{p_\text{n}(y|x)}\right)\right] \qquad (16)$$

with $f(z) := 1/(1 + 1/z)$. Using Jensen's inequality for the concave function $f$, we find that the right-hand side of Eq. 16 has the upper bound $f\left(\mathbb{E}_{p_\text{n}(y|x)}[p_\mathcal{D}(y|x)/p_\text{n}(y|x)]\right) = f(1) = \frac{1}{2}$, which it reaches precisely if the argument of $f$ in Eq. 16 is a constant, i.e., iff $p_\text{n}(y|x) = p_\mathcal{D}(y|x)\ \forall y \in \mathcal{Y}$. $\quad\square$

Table 1: Sizes of data sets and hyperparameters. $N$ = number of training points; $C$ = number of categories (after preprocessing); $\rho$ = learning rate; $\lambda$ = regularizer; $\sigma_0^2$ = prior variance.

| Data set | Size of data set | adv. neg. s. (proposed) | uniform neg. s. | $\propto$ freq. neg. s. | NCE | A&R | OVE |
|---|---|---|---|---|---|---|---|
| Wikipedia-500K | $N$=1,646,302 $C$=217,240 | $\rho$=0.01 $\lambda$=0.001 | $\rho$=0.001 $\lambda$=0.0001 | $\rho$=0.003 $\lambda$=$10^{-5}$ | $\rho$=0.01 $\lambda$=0.003 | $\rho$=0.03 $\sigma_0^2$=0.1 | $\rho$=0.02 $\sigma_0^2$=1 |
| Amazon-670K | $N$=490,449 $C$=213,874 | $\rho$=0.01 $\lambda$=0.001 | $\rho$=0.01 $\lambda$=0.0003 | $\rho$=0.003 $\lambda$=$10^{-5}$ | $\rho$=0.01 $\lambda$=0.001 | $\rho$=0.1 $\sigma_0^2$=10 | $\rho$=0.03 $\sigma_0^2$=10 |

## 5 RESULTS

We evaluated the proposed adversarial negative sampling method on two established benchmarks by comparing speed of convergence and predictive performance against five different baselines.

**Datasets, Preprocessing and Model.** We used the Wikipedia-500K and Amazon-670K data sets from the Extreme Classification Repository (Bhatia et al.) with $K = 512$-dimensional XML-CNN features (Liu et al., 2017) downloaded from (Saxena). As oth data sets contain multiple labels per data point we follow the approach in (Ruiz et al., 2018) and keep only the first label for each data point. Table 1 shows the resulting sizes. We fit a liner model with scores $\xi_y(x, \phi) = x^\top w_y + b_y$, where the model parameters $\phi$ are the weight vectors $w_y \in \mathbb{R}^K$ and biases $b_y \in \mathbb{R}$ for each label $y$.

**Baselines.** We compare our proposed method to five baselines: (i) standard negative sampling with a uniform noise distribution; (ii) negative sampling with an unconditional noise distribution $p_n(y')$ set to the empirical label frequencies; (iii) noise contrastive estimation (NCE, see below); (iv) 'Augment and Reduce' (Ruiz et al., 2018); and (v) 'One vs. Each' (Titsias, 2016). We do not compare to full softmax classification, which would be unfeasible on the large data sets (see Table 1; a single epoch of optimizing the full softmax loss would scale as $O(NCK)$). However, we provide additional results that compare softmax against negative sampling on a smaller data set in Appendix A.2.

NCE (Gutmann & Hyvärinen, 2010) is sometimes used as a synonym for negative sampling in the literature, but the original proposal of NCE is more general and allows for a nonuniform base distribution. We use our trained auxiliary model (Section 3) for the base distribution of NCE. Compared to our proposed method, NCE uses the base distribution only during training and not for predictions. Therefore, NCE has to re-learn everything that is already captured by the base distribution. This is less of an issue in the original setup for which NCE was proposed, namely unsupervised density estimation over a continuous space. By contrast, training a supervised classifier effectively means training a separate model for each label $y \in \mathcal{Y}$, which is expensive if $\mathcal{Y}$ is large. Thus, having to re-learn what the base distribution already captures is potentially wasteful.

**Hyperparameters.** We tuned the hyperparameters for each method individually using the validation set. Table 1 shows the resulting hyperparameters. For the proposed method and baselines (i)-(iii) we used an Adagrad optimizer (Duchi et al., 2011) and considered learning rates $\rho \in \{0.0003, 0.001, 0.003, 0.01, 0.03\}$ and regularizer strengths (see Eq. 6) $\lambda \in \{10^{-5}, 3 \times 10^{-5}, \ldots, 0.03\}$. For 'Augment and Reduce' and 'One vs. Each' we used the implementation published by the authors (Ruiz), and tuned the learning rate $\rho$ and prior variance $\sigma_0^2$. For the auxiliary model, we used a feature dimension of $k = 16$ and regularizer strength $\lambda_n = 0.1$ for both data sets.

**Results.** Figure 1 shows our results on the Wikipedia-500K data set (left two plots) and the Amazon-670K data set (right two plots). For each data set, we plot the the predictive log likelihood per test data point (first and third plot) and the predictive accuracy (second and fourth plot). The green curve in each plot shows our proposed adversarial negative sampling methods. Both our method and NCE (orange) start slightly shifted to the right to account for the time to fit the auxiliary model.

Our main observation is that the proposed method converges orders of magnitude faster and reaches better accuracies (second and third plot in Figure 1) than all baselines. On the (smaller) Amazon-670K data set, standard uniform and frequency based negative sampling reach a slightly higher predictive

Figure 1: Learning curves for our proposed adversarial negative sampling method (green) and for five different baselines on two large data sets (see Table 1).

log likelihood, but our method performs considerably better in terms of predictive accuracy on both data sets. This may be understood as the predictive accuracy is very sensitive to the precise scores of the highest ranked labels, as a small change in these scores can affect which label is ranked highest. With adversarial negative sampling, the training procedure focuses on getting the scores of the highest ranked labels right, thus improving in particular the predictive accuracy.

## 6 RELATED WORK

**Efficient Evaluation of the Softmax Loss Function.** Methods to speed up evaluation of Eq. 1 include augmenting the model by adding auxiliary latent variables that can be marginalized over analytically (Galy-Fajou et al., 2019; Wenzel et al., 2019; Ruiz et al., 2018; Titsias, 2016). More closely related to our work are methods based on negative sampling (Mnih & Hinton, 2009; Mikolov et al., 2013) and noise contrastive estimation (Gutmann & Hyvärinen, 2010). Generalizations of negative sampling to non-uniform noise distributions have been proposed, e.g., in (Zhang & Zweigenbaum, 2018; Chen et al., 2018; Wang et al., 2014; Gutmann & Hyvärinen, 2010). Our method differs from these proposals by drawing the negative samples from a conditional distribution that takes the input feature into account, and by requiring the model to learn only correlations that are not already captured by the noise distribution. We further derive the optimal distribution for negative samples, and we propose an efficient way to approximate it via an auxiliary model. Adversarial training (Miyato et al., 2017) is a popular method for training deep generative models (Tu, 2007; Goodfellow et al., 2014). By contrast, our method trains a discriminative model over a discrete set of labels (see also our comparison to GANs at the end of Section 2.2).

A different sampling-based approximation of softmax classification is 'sampled softmax' (Bengio et al., 2003). It directly approximates the sum over classes $y'$ in the loss (Eq. 1) by sampling, which is biased even for a uniform sampling distribution. A nonuniform sampling distribution can remove or reduce the bias (Bengio & Senécal, 2008; Blanc & Rendle, 2018; Rawat et al., 2019). By contrast, our method uses negative sampling, and it uses a nonuniform distribution to reduce the gradient variance.

**Decision Trees.** Decision trees (Somvanshi & Chavan, 2016) are popular in the extreme classification literature (Agrawal et al., 2013; Jain et al., 2016; Prabhu & Varma, 2014; Siblini et al., 2018; Weston et al., 2013; Bhatia et al., 2015; Jasinska et al., 2016). Our proposed method employs a probabilistic decision tree that is similar to Hierarchical Softmax (Morin & Bengio, 2005; Mikolov et al., 2013). While decision trees allow for efficient training and sampling in $O(\log C)$ time, their hierarchical architecture imposes a structural bias. Our proposed method trains a more expressive model without such a structural bias on top of the decision tree to correct for any structural bias.

## 7 CONCLUSIONS

We proposed a simple method to train a classifier over a large set of labels. Our method is based on a scalable approximation to the softmax loss function via a generalized form of negative sampling. By generating adversarial negative samples from an auxiliary model, we proved that we maximize the signal-to-noise ratio of the stochastic gradient estimate. We further show that, while the auxiliary model introduces a bias, we can remove the bias at test time. We believe that due to its simplicity, our method can be widely used, and we publish the code[1] of both the main and the auxiliary model.

---

[1] https://github.com/mandt-lab/adversarial-negative-sampling

ACKNOWLEDGEMENTS

Stephan Mandt acknowledges funding from DARPA (HR001119S0038), NSF (FW-HTF-RM), and Qualcomm.

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

APPENDIX

A.1  DETAILS OF THE PROOF OF THEOREM 2

In the nonparametric limit, the score functions $\xi_y(x, \phi)$ are so flexible that they can take arbitrary values for all $x$ in the data set and all $y \in \mathcal{Y}$. Taking advantage of the invariance of $\bar{\eta}$ under reparameterization, we parameterize the model directly by its scores. We use the shorthand $\xi_{x,y} \equiv \xi_y(x, \phi)$, and we denote the collection of all scores over all $x$ and $y \in \mathcal{Y}$ by boldface $\boldsymbol{\xi} \equiv (\xi_{x,y})_{x,y}$.

**Hessian.**  Eq. 2 defines the loss $\ell_{\text{neg.sampl.}}$ as a stochastic function. SGD minimizes its expectation,

$$\ell(\boldsymbol{\xi}) := \mathbb{E}\big[\ell_{\text{neg.sampl.}}(\boldsymbol{\xi})\big] = \sum_x \sum_{y \in \mathcal{Y}} \big[ -p_{\mathcal{D}}(y|x) \log \sigma(\xi_{x,y}) - p_{\text{n}}(y|x) \log \sigma(-\xi_{x,y}) \big] \quad \text{(A1)}$$

where the sum over $x$ runs over all feature vectors in the training set. We obtain the gradient

$$g_{x,y} \equiv \nabla_{\xi_{x,y}} \ell(\boldsymbol{\xi}) = -p_{\mathcal{D}}(y|x) \sigma(-\xi_{x,y}) + p_{\text{n}}(y|x) \sigma(\xi_{x,y}) \quad \text{(A2)}$$

where we used the relation $\nabla_z \log \sigma(z) = \sigma(-z)$. The gradient is a vector whose components span all combinations of $x$ and $y$. The Hessian matrix $H_\ell$ contains the derivatives of each gradient component $g_{x,y}$ by each coordinate $\xi_{\tilde{x},\tilde{y}}$. Since $g_{x,y}$ in Eq. A2 depends only on the single coordinate $\xi_{x,y}$, only the diagonal parts of the Hessian are nonzero, i.e., the components with $x = \tilde{x}$ and $y = \tilde{y}$. Thus,

$$H_\ell = \text{diag}(\alpha_{x,y}) \qquad \text{with} \qquad \alpha_{x,y} = \nabla_{\xi_{x,y}} g_{x,y}. \quad \text{(A3)}$$

Using the identity $\nabla_z \sigma(z) = \sigma(z) \sigma(-z)$, we find

$$\alpha_{x,y} = \big[ p_{\mathcal{D}}(y|x) + p_{\text{n}}(y|x) \big] \sigma(-\xi_{x,y}) \sigma(\xi_{x,y}). \quad \text{(A4)}$$

Since we evaluate the Hessian in the nonparametric limit at the minimum of the loss, the scores $\xi_{x,y}$ satisfy Eq. 11, i.e.,

$$p_{\mathcal{D}}(y|x) \sigma(-\xi_{x,y}) = p_{\text{n}}(y|x) \sigma(\xi_{x,y}). \quad \text{(A5)}$$

This allows us to simplify Eq. A4 by eliminating $p_{\mathcal{D}}$,

$$\alpha_{x,y} = p_{\text{n}}(y|x) \underbrace{\big[\sigma(\xi_{x,y}) + \sigma(-\xi_{x,y})\big]}_{=1} \sigma(\xi_{x,y}) = p_{\text{n}}(y|x) \sigma(\xi_{x,y}). \quad \text{(A6)}$$

Eqs. A3 and A6 together prove Eq. 13 of the main text.

**Noise Covariance Matrix.**  SGD uses estimates $\hat{\ell}$ of the loss function in Eq. A1, obtained by drawing a positive sample $(x, y) \sim \mathcal{D}$ and a label for the negative sample $y' \sim p_{\text{n}}(y'|x)$, thus

$$\hat{\ell}(\boldsymbol{\xi}) = -N \big[ \log \sigma(\xi_{x,y}) + \log \sigma(-\xi_{x,y'}) \big] \quad \text{(A7)}$$

where the factor of $N \equiv |\mathcal{D}|$ is because the sum over $x$ in Eq. A1 scales proportionally to the size of the data set $\mathcal{D}$ (in practice one typically normalizes the loss function by $N$ without affecting the signal to noise ratio). One uses $\hat{\ell}$ to obtain unbiased gradient estimates $\hat{g}$. We introduce new symbols $\tilde{x}$ and $\tilde{y}$ for the components $\hat{g}_{\tilde{x},\tilde{y}}$ of the gradient estimate to avoid confusion with the $x$ and $y$ drawn from the data set and the $y'$ drawn from the noise distribution in Eq. A7 above. Since the scores are independent variables in the nonparametric limit, the derivative $\nabla_{\xi_{\tilde{x},\tilde{y}}} \xi_{x,y}$ is one if $\tilde{x} = x$ and $\tilde{y} = y$, and zero otherwise. We denote this by indicator functions $\mathbf{1}_{\tilde{x}=x}$ and $\mathbf{1}_{\tilde{y}=y}$. Thus, we obtain

$$\hat{g}_{\tilde{x},\tilde{y}} \equiv \nabla_{\xi_{\tilde{x},\tilde{y}}} \hat{\ell}(\boldsymbol{\xi}) = -N \big[ \sigma(-\xi_{x,y}) \mathbf{1}_{\tilde{y}=y} - \sigma(\xi_{x,y'}) \mathbf{1}_{\tilde{y}=y'} \big] \mathbf{1}_{\tilde{x}=x} \quad \text{(A8)}$$

We evaluate the covariance matrix of $\hat{g}$ at the minimum of the loss function. Here, $\mathbb{E}[\hat{g}] \equiv g = 0$, and thus $\text{Cov}[\hat{g}, \hat{g}] \equiv \mathbb{E}[\hat{g}\,\hat{g}^\top] - \mathbb{E}[\hat{g}]\,\mathbb{E}[\hat{g}^\top]$ simplifies to $\mathbb{E}[\hat{g}\,\hat{g}^\top]$. Introducing yet another pair of indices $\tilde{\tilde{x}}$ and $\tilde{\tilde{y}}$ to distinguish the two factors of $\hat{g}$, we denote the components of the covariance matrix as

$$\text{Cov}[\hat{g}_{\tilde{x},\tilde{y}}, \hat{g}_{\tilde{\tilde{x}},\tilde{\tilde{y}}}] \equiv \mathbb{E}_{(x,y)\sim\mathcal{D}, \, y'\sim p_{\text{n}}} \big[ \hat{g}_{\tilde{x},\tilde{y}} \, \hat{g}_{\tilde{\tilde{x}},\tilde{\tilde{y}}} \big]. \quad \text{(A9)}$$

Here, the expectation is over $p_{\mathcal{D}}(x, y) \, p_{\text{n}}(y'|x) = p_{\mathcal{D}}(x) \, p_{\mathcal{D}}(y|x) \, p_{\text{n}}(y'|x)$. We start with the evaluation of the expectation over $x$, using $\mathbb{E}_{x \sim p_{\mathcal{D}}}[\,\cdot\,] = \frac{1}{N} \sum_x [\,\cdot\,]$ where the sum runs over all $x$ in the

data set. If $x \neq \tilde{x}$ or $x \neq \tilde{\tilde{x}}$, then either one of the two gradient estimates $\hat{g}$ in the expectation on the right-hand side of Eq. A9 vanishes. Therefore, only terms with $x = \tilde{x} = \tilde{\tilde{x}}$ contribute, and the covariance matrix is block diagonal in $x$ as claimed in Eq. 14 of the main text. The blocks $C_x$ of the block diagonal matrix have entries

$$(C_x)_{\tilde{y},\tilde{\tilde{y}}} \equiv \mathrm{Cov}[\hat{g}_{x,\tilde{y}}, \hat{g}_{x,\tilde{\tilde{y}}}] = \frac{1}{N} \, \mathbb{E}_{p_{\mathcal{D}}(y|x) \, p_{\mathrm{n}}(y'|x)} \big[ \hat{g}_{x,\tilde{y}} \, \hat{g}_{x,\tilde{\tilde{y}}} \big]. \tag{A10}$$

where we find for the product $\hat{g}_{x,\tilde{y}} \, \hat{g}_{x,\tilde{\tilde{y}}}$ by inserting Eq. A8 and multiplying out the terms,

$$\begin{aligned} \hat{g}_{x,\tilde{y}} \, \hat{g}_{x,\tilde{\tilde{y}}} = N^2 \Big[ &\Big( \sigma(-\xi_{x,\tilde{y}})^2 \, \mathbf{1}_{\tilde{y}=y} + \sigma(\xi_{x,\tilde{y}})^2 \, \mathbf{1}_{\tilde{y}=y'} \Big) \mathbf{1}_{\tilde{y}=\tilde{\tilde{y}}} \\ &- \sigma(-\xi_{x,\tilde{y}}) \, \sigma(\xi_{x,\tilde{\tilde{y}}}) \, \mathbf{1}_{\tilde{y}=y \, \wedge \, \tilde{\tilde{y}}=y'} - \sigma(\xi_{x,\tilde{y}}) \, \sigma(-\xi_{x,\tilde{\tilde{y}}}) \, \mathbf{1}_{\tilde{y}=y' \, \wedge \, \tilde{\tilde{y}}=y} \Big] \end{aligned} \tag{A11}$$

Taking the expectation in Eq. A10 leads to the following substitutions:

$$\mathbf{1}_{\tilde{y}=y} \longrightarrow p_{\mathcal{D}}(\tilde{y}|x); \quad \mathbf{1}_{\tilde{\tilde{y}}=y} \longrightarrow p_{\mathcal{D}}(\tilde{\tilde{y}}|x); \quad \mathbf{1}_{\tilde{y}=y'} \longrightarrow p_{\mathrm{n}}(\tilde{y}|x); \quad \mathbf{1}_{\tilde{\tilde{y}}=y'} \longrightarrow p_{\mathrm{n}}(\tilde{\tilde{y}}|x). \tag{A12}$$

Thus, we find,

$$\begin{aligned} (C_x)_{\tilde{y},\tilde{\tilde{y}}} = N \Big[ &\Big( p_{\mathcal{D}}(\tilde{y}|x) \, \sigma(-\xi_{x,\tilde{y}})^2 + p_{\mathrm{n}}(\tilde{y}|x) \, \sigma(\xi_{x,\tilde{y}})^2 \Big) \mathbf{1}_{\tilde{y}=\tilde{\tilde{y}}} \\ &- p_{\mathcal{D}}(\tilde{y}|x) \, p_{\mathrm{n}}(\tilde{\tilde{y}}|x) \, \sigma(-\xi_{x,\tilde{y}}) \, \sigma(\xi_{x,\tilde{\tilde{y}}}) - p_{\mathrm{n}}(\tilde{y}|x) \, p_{\mathcal{D}}(\tilde{\tilde{y}}|x) \, \sigma(\xi_{x,\tilde{y}}) \, \sigma(-\xi_{x,\tilde{\tilde{y}}}) \Big]. \end{aligned} \tag{A13}$$

Using Eq. A5, we can again eliminate $p_{\mathcal{D}}$,

$$\begin{aligned} (C_x)_{\tilde{y},\tilde{\tilde{y}}} &= N \Big[ \Big( p_{\mathrm{n}}(\tilde{y}|x) \, \sigma(-\xi_{x,\tilde{y}}) \, \sigma(\xi_{x,\tilde{y}}) + p_{\mathrm{n}}(\tilde{y}|x) \, \sigma(\xi_{x,\tilde{y}})^2 \Big) \mathbf{1}_{\tilde{y}=\tilde{\tilde{y}}} \\ &\qquad - 2 p_{\mathrm{n}}(\tilde{y}|x) \, p_{\mathrm{n}}(\tilde{\tilde{y}}|x) \, \sigma(\xi_{x,\tilde{y}}) \, \sigma(\xi_{x,\tilde{\tilde{y}}}) \Big] \\ &= N \Big[ \alpha_{x,\tilde{y}} \underbrace{\Big( \sigma(-\xi_{x,\tilde{y}}) + \sigma(\xi_{x,\tilde{y}}) \Big)}_{=1} \mathbf{1}_{\tilde{y}=\tilde{\tilde{y}}} - 2 \, \alpha_{x,\tilde{y}} \, \alpha_{x,\tilde{\tilde{y}}} \\ &= N \big[ \alpha_{x,\tilde{y}} \, \mathbf{1}_{\tilde{y}=\tilde{\tilde{y}}} - 2 \, \alpha_{x,\tilde{y}} \, \alpha_{x,\tilde{\tilde{y}}} \big]. \end{aligned} \tag{A14}$$

Eq. A14 is the component-wise explicit form of Eq. 14 of the main text.

## A.2   Experimental Comparison Between Softmax Classification and Negative Sampling

We provide additional experimental results that evaluate the performance gap due to negative sampling compared to full softmax classification on a smaller data set. Theorem 1 states an equivalence between negative sampling and softmax classification. However, this equivalence strictly holds only (i) in the nonparametric limit, (ii) without regularization, and (iii) if the optimizer really finds the global minimum of the loss function. In practice, all three assumptions hold only approximately.

**Data Set and Preprocessing.**   To evaluate the performance gap experimentally, we used "EURLex-4K" data set (Bhatia et al.; Mencia & Fürnkranz, 2008), which is small enough to admit direct optimization of the softmax loss function. Similar to the preprocessing of the two main data sets described in Section 5 of the main text, we converted the multi-class classification problem into a single-class classification problem by selecting the label with the smallest ID for each data point, and discarding any data points without any labels. We split off $10\%$ of the training set for validation, and report results on the provided test set. This resulted in a training set with $N = 13,960$ data points and $C = 3,687$ categories. As in the main paper, we reduced the feature dimension to $K = 512$ (using PCA for simplicity here).

**Model and Hyperparameters.**   The goal of these experiments is to evaluate the performance gap due to negative sampling in general. We therefore fitted the same affine linear model as described in Section 5 of the main text using the full softmax loss function (Eq. 1) and the simplest form of negative sampling (Eq. 2), i.e., negative sampling with a uniform noise distribution. We added a quadratic regularizer with strength $\lambda$ to both loss functions.

For both methods, we tested the same hyperparameter combinations as in Section 5 on the validation set using early stopping. For softmax, we extended the range of tested learning rates up to $\rho = 10$ as higher learning rates turned out to perform better in this method (this can be understood due to the low gradient noise). The optimal hyperparameters for softmax turned out to be a learning rate of $\rho = 0.3$ and regularization strength $\lambda = 3 \times 10^{-4}$. For negative sampling, we found $\rho = 3 \times 10^{-3}$ and $\lambda = 3 \times 10^{-4}$.

**Results.** We evaluated the predictive accuracy for both methods. With the full softmax method, we obtain $33.6\%$ correct predictions on the test set, whereas the predictive accuracy drops to $26.4\%$ with negative sampling. This suggests that, when possible, minimizing the full softmax loss function should be preferred. However, in many cases, the softmax loss function is too expensive.

