# OpenReview forum: "Extreme Classification via Adversarial Softmax Approximation"
_ICLR.cc/2020/Conference — Accept (Poster)_

### Official Review · AnonReviewer3 · 2019-10-23
**Official Blind Review #3**

**Rating:** 3

**Review:**

The paper presents a method for negative sampling for softmax when dealing with classification of data to one from a large number of classes. Its main idea is to negative sample those classes which lead to higher signal to noise ratio than for uniform negative sampling. This is based on building an auxilary model using decision tree from which the adversarial negative classes are sampled, so that the distribution of the negative samples can be close to the positive ones leading to higher SNR while training. The proposed method is compared to other methods for negative samping on two publicly available large-scale datasets from the extreme classification with XML-XNN features.

Positives :
1. The proposed approach with adversarial negative sampling using an auxilary model seems interesting
2. It scales well to datasets with large number of classes.

Negatives :
The experimental evaluation of the proposed approach lacks completeness and does not look convincing for the following reasons :
1. It misses out a recent state-of-the-art method (Slice) for negative sampling on same datasets [1], which also addresses the same problem of sampling most promising negative classes but in a different way. Furthermore, [1] also compares against many other sota methods missed out in this paper on the many other datasets datasets including those in this paper but in a more general multi-label setting.
2. The paper only compares against other negative sampling approaches such as AandR, NCE, and does not show what happens when no negative sampling is done such as done in (DiSMEC) [2]. This is important to understand what (if at all) is lost by doing approximation as proposed. For instance, a quick experiment reveals that DiSMEC can give about 19% accuracy on Wiki500 dataset, which is better than that achieved by the proposed method. Though it is computationally expensive but due to its simplicity, it must be discussed nevertheless to give a complete picture.
Instead the OVE baseline used in the paper seems quite sub-optimal in the first place, and hence stronger baselines [1,2] for which the code and results are readily available and have been duly tested in the community must be used and discussed.

Another aspect that the paper misses out is the role of fat-tailed distribution [3,4] of the instances among labels, which is a property of typical datasets in this regime. It is possible that one can get good accuracy but poor performance on tail-labels due to approaximations. The performance on tail-labels on appropriate metrics other than accuracy, such as MacroF1, should be evaluated.

Also, the proposed approach must be tested on more datasets including the smaller ones such as EURLex (also used in works referenced in the paper) on which it is easier to compare with other methods (such as DiSMEC, Slice and AttentionXML [5]) without encountering computational constraints and also bigger ones such as Amazon3M, also avilable from the repository.

Finally, it must be investigated if the proposed method can be extended to the multi-label setting or are there inherent limitations of the model in this setting. The possibility to extend it to the general multi-label setting would make this approach more promising and directly comparable to wide range of algorithms.

[1] H. Jain,  V. Balasubramanian,  B. Chunduri and M. Varma, Slice: Scalable linear extreme classifiers trained on 100 million labels for related searches, in WSDM 2019.
[2] R. Babbar, and B. Schölkopf, DiSMEC - Distributed Sparse Machines for Extreme Multi-label Classification in WSDM, 2017.
[3] H. Jain, Y. Prabhu, and M. Varma, Extreme Multi-label Loss Functions for Recommendation, Tagging, Ranking & Other Missing Label Applications in KDD, 2016.
[4] R. Babbar, and B. Schölkopf, Data Scarcity, Robustness and Extreme Multi-label Classification in Machine Learning Journal and European Conference on Machine Learning, 2019.
[5] AttentionXML: Extreme Multi-Label Text Classification with Multi-Label Attention Based Recurrent Neural Networks, NIPS 2019

**Experience Assessment:**

I have published in this field for several years.

**Review Assessment: Checking Correctness Of Derivations And Theory:**

I assessed the sensibility of the derivations and theory.

**Review Assessment: Checking Correctness Of Experiments:**

I carefully checked the experiments.

**Review Assessment: Thoroughness In Paper Reading:**

I read the paper at least twice and used my best judgement in assessing the paper.

---

> ### Author Response · Authors · 2019-11-15
> **Clarification of Our Paper's Focus and Our Contributions**
>
> We thank the reviewer for pointing us to an extensive list of literature. We admit that related work in the wider field of extreme classification is not sufficiently acknowledged in the Related Work section of our paper and we will add a discussion that includes the suggested references in the final version of our paper. We would like to clarify that our paper is about negative sampling and we would like to stress the contributions in this context.
>
> First, we believe that our paper significantly advances the theoretical understanding of negative sampling, with practical consequences. We would like to point out in particular Theorem 2. To the best of our knowledge, this is the first formalization and rigorous proof of the intuition that “hard” negative samples are “better”. While this intuition has been invoked in the literature before, the notions of “hard” and “better” are usually somewhat fuzzy. Our paper formalizes this intuition by defining a well-motivated scalar measure of the signal-to-noise ratio and proving rigorously that this ratio is optimal for negative samples that are “hard” in a well-defined way. This theoretical insight has practical consequences as it allowed us to design a very simple yet effective way to generate near-optimal negative samples.
>
> Second, we provide experimental results on two established benchmarks and compare against five baselines. We agree that three of our baselines are variants of negative sampling. This is because our paper proposes a simple improvement of negative sampling.
> Negative sampling is a very popular method in practice due to its simplicity (for example, it is very common in the knowledge graph embedding literature).
>
> Finally, the auxiliary model proposed in our paper has merit on its own. It is a simple model that can be fitted deterministically and highly efficiently, requiring tuning of only a single model hyperparameter and no hyperparameters for the training schedule.
>
> Given these contributions, the paper’s focus on negative sampling, and the already extensive baseline comparisons, we respectfully disagree with the reviewer’s request for more comparisons to non-negative-sampling methods. We acknowledge that the extreme classification community has developed many algorithms with high predictive accuracy. We hope that the community will appreciate that our paper proposes a very simple, theoretically well-founded, and (as the reviewer acknowledges) faster alternative that significantly outperforms very popular approaches similar to it. We also believe that single-label classification is of enormous practical relevance, and that our proposal should not be dismissed on the grounds that it cannot also perform multi-label classification.
>
> However, we would like to thank the reviewer for their idea to run experiments on the smaller EURLex data set. Although negative sampling in this regime is somewhat artificial, a smaller data set allows us to compare against full softmax classification, providing insight into the approximation gap due to negative sampling in general. We are running these experiments and will report results in the final version of the paper. We apologize that we could not finish these experiments in time for the rebuttal deadline.

---

### Official Review · AnonReviewer1 · 2019-10-23
**Official Blind Review #1**

**Rating:** 6

**Review:**

This paper focuses on efficient and fast training in the extreme classification setting where the number of classes C is very large. In this setting, naively using softmax based loss function incurs a prohibitively large cost as the cost of computing the loss value for each example scales linearly with C. One way to circumvent this issue is to only utilize a small subset of negative classes during the loss computation. However, uniformly sampling this subset from all the negative classes suffers from the slow convergence as such sampled negatives are not very informative for the underlying classification task.

The paper proposes a method to sample the negatives in a non-uniform manner. In particular, given an example, an adversarial auxiliary model that is tasked with tracking the data distribution samples the hardest (adversarial) negatives for the example. The proposed method to sample negatives has a computational cost log(C) and reduces the noise in the gradient. The authors then demonstrate the utility of their proposed approach on two well-established extreme classification datasets, i.e., Wikipedia-500K and Amazon-670K. The proposed method shows improvement over some natural baselines in terms of the wall-time for the convergence of the training process.

Comments

1. The paper has some nice contributions and discusses the key ideas in reasonable detail. However, the reviewer feels that the authors gloss over many relevant prior works and fail to put their results in the right context. There has been quite a bit of work on non-uniformly sampling "hard" negative classes. For example, see [1], [2], [3], [4]. In fact, [3] and [4] propose methods to sample negatives from a distribution that closely approximates the softmax distribution at the cost that scales logarithmically in C, essentially providing the hard negative without having to keep an auxiliary model. Can the authors discuss their work in the context of these works?

[1] Reddi et al., Stochastic Negative Mining for Learning with Large Output Spaces.
[2] Grave et al., Efficient softmax approximation for GPUs.
[3] Blanc and Rendle, Adaptive Sampled Softmax with Kernel-based Sampling.
[4] Rawat et al., Sampled Softmax with Random Fourier Features.

2. In experiments, the authors do not include the performance of softmax loss (eq. (1)) due to its large computational cost. However, it would be nice to compare the proposed method with eq. (1) at least for slightly smaller datasets from the extreme classification repository.

3. In Sec. 4, "We formalize and proof..." --> "We formalize and prove..."

4. In Sec. 1, "We present experiments on several two classifications..." ---> "We present experiments on two classifications..."

5. Table 1 seems to have some typos. E.g., N is the same for both the data sets. Please fix these issues.

**Experience Assessment:**

I have published one or two papers in this area.

**Review Assessment: Checking Correctness Of Derivations And Theory:**

I assessed the sensibility of the derivations and theory.

**Review Assessment: Checking Correctness Of Experiments:**

I carefully checked the experiments.

**Review Assessment: Thoroughness In Paper Reading:**

I read the paper thoroughly.

---

> ### Author Response · Authors · 2019-11-15
> **"Negative Sampling" vs. "Sampled Softmax"**
>
> We thank the reviewer for pointing us to relevant additional literature. While we believe that there is a confusion between “negative sampling” (used in our paper) and “sampled softmax” (used in [3] and [4]), see below, we still find the references relevant and we uploaded a new version of our paper that discusses them in the Related Works section. The updated version also fixes the grammar errors and issues with Table 1 kindly pointed out by the reviewer.
>
> We like the reviewer’s idea of evaluating our method on smaller data sets where the full softmax loss can be optimized. Although negative sampling in this regime feels somewhat artificial and we don’t expect it to perform as well as real softmax classification, such experiments provide insight into the approximation gap due to negative sampling in general. We are running experiments on the smaller EURLex data set and will report results in the final version of the paper. We apologize that we could not finish these experiments in time for the rebuttal deadline.
>
> Before addressing the issue of “negative sampling” vs. “sampled softmax” (Refs. [3] and [4] in the review), we would like to stress the theoretical contributions of our paper, in particular Theorem 2. To the best of our knowledge, our paper provides the first formalization and rigorous proof of the intuition that “hard” negative samples are “better”. While this intuition has been invoked in the literature before, the notions of “hard” and “better” are usually somewhat fuzzy. Our paper formalizes this intuition by defining a well-motivated scalar measure of the signal-to-noise ratio and proving rigorously that this ratio is optimal for negative samples that are “hard” in a well-defined way. This theoretical insight has practical consequences as it allowed us to design a very simple yet effective way to generate near-optimal negative samples.
>
>
> NEGATIVE SAMPLING VS. SAMPLED SOFTMAX
>
> References [3] and [4] in the review both discuss nonuniform sampling for a method called “sampled softmax”, which is related but different from negative sampling. The main difference is that “sampled softmax” is biased even under a uniform distribution, whereas “negative sampling” with a uniform noise distribution is unbiased. References [3] and [4] thus use a nonuniform sampling distribution for bias reduction whereas our paper uses it for variance reduction.
>
> In detail, sampled softmax directly approximates the sum over all classes in the softmax loss function (Eq. 1 in our paper) by sampling. This introduces a bias since the sum appears inside the logarithm, which is nonlinear. By contrast, negative sampling does not directly approximate the softmax loss function. Instead, it estimates a different loss function, namely for binary classification (Eq. 2 in our paper). Although the loss function is very different, minimizing it yields the same trained model parameters as minimizing the softmax loss function, as we show in Theorem 1 (in the nonparametric limit). In this sense, negative sampling approximates softmax classification.
>
>
> > In fact, [3] and [4] propose methods to sample negatives from a distribution that closely
> > approximates the softmax distribution [...] essentially providing the hard negative [...]
>
> To our understanding, [3] and [4] do not generate “hard” negative samples, i.e., negative samples that resemble positive samples from the data distribution. Their sampling distributions are designed to approximate the model distribution, not the data distribution, which is very different at the beginning of training. Also, their use of a nonuniform sampling distribution is not a means to speed up convergence by reducing gradient noise. It is simply a necessity to make the sampled softmax approximation unbiased (see Theorem 2.1 in [3] and end of Section 2 in [4]).
>
> > [...] without having to keep an auxiliary model.
>
> While the authors do not refer to it as an “auxiliary model”, the “summary vector” z in Eq. 8 of [3] serves an equivalent purpose. The vector even has to be updated in an expensive operation during training of the main model because the sampling distribution in sampled softmax has to follow the (changing) model distribution. By contrast, our auxiliary model can be kept static, thus simplifying training and leading to a well-defined (static) loss function.
>
>
> CONCERNING REFERENCES  [1] and [2]
>
> Refs. [1] and [2] are orthogonal to our approach: [1] generalizes negative sampling to a “top-k” ranking task but uses only uniform sampling; [2] does not use sampling as far as we can tell. It instead focuses on a deterministic approximation of the softmax loss that is engineered for a computational model of a GPU.

---

### Official Review · AnonReviewer2 · 2019-11-01
**Official Blind Review #2**

**Rating:** 8

**Review:**

This work addresses the problem of training softmax classifiers when the number of classes is extreme. The authors improve the negative sampling method which is based on reducing the multi-class problem to a binary class problem by introducing randomly chosen labels in the training.  Their idea is generating the fake labels nonuniformly from an adversarial model (a decision tree). They show convincing results of improved learning rate.
The work is very technical in nature, but the proposal is presented in detail and in a didactic way with appropriate connections to alternative methods, so that it may be useful for the non-expert (as me).
That is the reason why I recommend to accept this work: even not being an expert I found the paper educative in introducing the problem and interesting in explaining the proposal.

**Experience Assessment:**

I have read many papers in this area.

**Review Assessment: Checking Correctness Of Derivations And Theory:**

I assessed the sensibility of the derivations and theory.

**Review Assessment: Checking Correctness Of Experiments:**

I assessed the sensibility of the experiments.

**Review Assessment: Thoroughness In Paper Reading:**

I read the paper at least twice and used my best judgement in assessing the paper.

---

> ### Author Response · Authors · 2019-11-15
> **Thank you for the positive review**
>
> We thank the reviewer for their favorable review. We are happy to hear that our derivations are comprehensible to a (self-proclaimed) non-expert. We believe that the approachability is also a strength of the proposed method: due to its simplicity, the method can be used as a drop-in replacement for the widely used negative sampling approach.

---

### Public Comment · ~Gladis_Ne_Limes1 · 2023-11-13
**re**

With a dedicated team or agency, website development projects can progress more rapidly https://mlsdev.com/blog/how-to-outsource-web-development . This is especially beneficial for businesses with tight deadlines or those looking to launch their online presence quickly. Scalability: Outsourcing offers scalability, allowing businesses to scale up or down based on project requirements. This flexibility is particularly valuable for companies with fluctuating development needs. Global Perspective: Outsourcing provides access to a global perspective, bringing in diverse ideas and insights. This can contribute to a more innovative and well-rounded website development process.

---

### Decision · Program_Chairs · 2019-12-19

**Decision:**

Accept (Poster)

**Comment:**

The paper proposes a fast training method for extreme classification problems where number of classes is very large. The method improves the negative sampling (method which uses uniform distribution to sample the negatives) by using an adversarial auxiliary model to sample negatives in a non-uniform manner. This has logarithmic computational cost and minimizes the variance in the gradients. There were some concerns about missing empirical comparisons with methods that use sampled-softmax approach for extreme classification. While these comparisons will certainly add further value to the paper, the improvement over widely used method of negative sampling and a formal analysis of improvement from hard negatives is a valuable contribution in itself that will be of interest to the community. Authors should include the experiments on small datasets to quantify the approximation gap due to negative sampling compared to full softmax, as promised.